# Composition of Phenolic Compounds in South African *Schinus molle* L. Berries

**DOI:** 10.3390/foods11101376

**Published:** 2022-05-10

**Authors:** Callistus Bvenura, Learnmore Kambizi

**Affiliations:** Department of Horticulture, Faculty of Applied Sciences, Cape Peninsula University of Technology, Bellville 7535, South Africa; callistusb@gmail.com

**Keywords:** flavonoids, phenolic acids, pseudo-spices, *Schinus molle*, tannins, underutilised indigenous foods

## Abstract

The *Schinus molle* tree is notoriously invasive in most parts of the world, and yet as a pseudospice, its berries potentially possess some significant health benefits which need to be explored. Therefore, polar metabolome of seed + husks (SH), husks (H), and de-hulled (DH) berries were profiled and quantified by untargeted metabolomics approach using UPLC-QTOF-MS. A total of 13 gallotannins, three phenolic acids, a phenolic acid glucoside, three phenolic acid esters, an organic acid, a gallotannin derivative, and nine flavonoids were detected and quantified. Phenolic acids ranged between 12.2–295.7; 4.9–77; and 89.7–1613.1 mg/kg in SH, DH seeds and H respectively. Flavonoids ranged between 1.8–267.5; 73.4–80.4; and 124–564.3 mg/kg in SH, DH seeds and H respectively. Gallotannins ranged between 1.1–146.6; 14.8–21.8; and 48.1–664.8 mg/kg in SH, DH seeds and H respectively. Feruloyltartaric A, quercetin 3-O-glucuronide, catechin digalloylshikimic acid B as well as digalloyl quinic acid were some of the dominant secondary metabolites revealed. These results indicate that *S. molle* berries are a rich source of secondary metabolites with elevated concentrations in the husks, while DH seeds possess lower concentrations to none. These findings open important insights into the potential of *S. molle* berries as a natural source of antioxidants for the food and pharmaceutical industries.

## 1. Introduction

Urbanisation and globalisation impacts including the proliferation of unhealthy foods, poverty and ultimately poor diets are the leading causes of NCDs especially in poor countries [1]. In fact, diets that are lacking in vegetables and fruits, seeds, nuts, omega-3 fatty acid rich foods and whole grains are the diet related driving forces [1]. Diets lacking these components are normally high in trans-fatty acids, sugar and sodium/salt which are detrimental to health [2]. The annual costs of treating and managing NCDs in South Africa are about US$34.2 billion and this is 10% of South Africa’s Gross Domestic Product [3]. A change in diet is recommended by the WHO [4]. This includes a massive reduction in salt intake, increasing diets that are high in antioxidants, and polyphenolic compounds as well as increased physical activity. The replacement of salt with spices could potentially reduce the salt intake while increasing the nutritional and polyphenolic compounds of the diet [5].

Non-communicable diseases (NCDs) including chronic respiratory diseases, cancer, diabetes, and cardiovascular diseases are the current leading cause of mortality worldwide [6]. These diseases account for 70% (40 million people) of global deaths [6]. In developing countries, these deaths account for about 50% of people under the age of 70 [7]. This data is disturbing because it shows the failure of developing countries to shake off the impact of NCDs which have previously been known as the developed countries’ curse. The World Health Organization (WHO) [4] estimated that by 2023, the NCD burden would have increased by 17% globally and 27% in Africa. The African continent is one of the poorest regions in the world. In South Africa, NCDs account for 51% of premature deaths as follows: cardiovascular diseases (19%), cancer (10%), diabetes (7%), chronic respiratory diseases (4%) and 11% for other NCDs [8]. 

Culinary herbs and spices have been used from time immemorial to enhance flavour in food and/or for their nutritional and health benefits [5]. However, indigenous/ pseudo-spices are generally neglected and underutilised and yet they have the potential to compliment the diet and improve its nutritional and health benefits. The effects of NCDs can potentially be ameliorated as well. If these pseudo-spices are neglected, their potential health benefits may never be realised. *Schinus molle* L. berries are a typical pseudo spice which needs to be investigated. *S. molle* is an Andean tree species from the Anacardiaceae family. This family comprises 83 genera and 860 known species [9]. The genus Schinus itself comprises about 30 known species [10]. This is native to South America and has several English names such as pink peppercorns, pepper tree, California pepper tree, peppercorn tree, pepperina, Peruvian pepper tree, and Peruvian mastic tree, among others [11]. This tree has now spread around the globe to the Mediterranean, tropical and subtropical regions including South Africa. Although this species is notoriously invasive in the semi-arid savanna of South Africa, some beneficial culinary uses of the berries have been reported especially in its native habitat. Besides substituting black pepper corns, some numerous uses have been reported, especially in the Andean region. According to García and Ormazabal [11], the whole berries are used to flavour drinks and syrups; in Mexico, an alcoholic fermented beverage known as ‘copalote’ or ‘copalocle’ is produced, in Californian they are cooked with vegetables or used as a garnish because of their mild, sweet taste that is also aromatic, citric, fruity, and floral. The dry whole berries are also mixed with butter and asparagus to prepare fish. A vinaigrette can also be derived while desserts and biscuits can be prepared using these berries. A drink known as ‘chicha’ or ‘molle’ in Peru is extracted from the whole berries for festivities [12]. These berries are generally not known in South Africa for these culinary benefits and therefore their potential nutritional and therapeutic role. The folkloric uses of the whole berries include uses as a mouth wash, sedative, purging agent, diuretic, disinfectant, analgesic, sciatica, antispasmodic, anti-septic, and hypotensor [13]. Furthermore, the berries reportedly possess anti-inflammatory properties and can cure rheumatic diseases, bronchitis, asthma, stomach and liver complaints, swelling of legs and arms as well as wound healing [13]. 

Numerous biological activities of the whole berries have also been reported against some microorganisms and pests [13,14,15]. The studies of [16,17,18,19] demonstrate the antioxidant, anti-inflammatory and antimicrobial activity of the berries. Perhaps these activities are strongly related to the polyphenolic compositions of the leaves [20,21,22,23,24,25], bark [26], as well as leaves and berries [17,19,27,28,29] of the *S. molle* tree. Fewer studies of the berries have been conducted. Specifically, refs. [16,30] reported the essential oil extract compounds while [31] and [32] reported these compounds in alcoholic extracts. Studies on the potential of *S. molle* berries as a spice and their associated pharmacological properties as well as phytochemistry and ethnobotany in South Africa are scarce. The full potential of this pseudo-spice needs to be explored and its full potential documented in South Africa. The cultivars that are actively growing also need to be investigated. The present study was therefore conducted to profile and quantify the secondary metabolites present in the ripe and mature berries of the species growing in the Western Cape province of South Africa.

## 2. Materials and Methods

### 2.1. Chemicals 

All the solvents and chemicals that were used, including Tetrafluoro acetic acid and formic acid were of LC-MS/analytical grade. From Millipore water purification system, ultrapure water with 18.2 MΩ cm^−1^ at 25 °C resistivity was used. Catechin standards, trifluoroacetic acetic acid, formic acid, methanol, and sodium formate were purchased from Sigma Chemicals, Johannesburg, South Africa.

### 2.2. Plant Materials 

Ripe and mature *S. molle* berries were collected from trees growing around Somerset West (34°05′ S 18°51′ E), Cape Town, South Africa in the month of April and May 2021 (Figure 1). The samples were put in sampling bags and immediately transported to the Cape Peninsula University of Technology laboratory for further processing. On arrival, the samples were separated into husks (H) and dehulled seeds (DH) while some were left in their original state, i.e., seed + husks (SH). The samples were dried at 50 °C in a dust free forced-draft oven to a constant weight. Samples were then finely ground and stored in glass vials in a refrigerator at 4 °C prior to further processing.

### 2.3. Sample Methanolic Extraction 

About 1 g of dry and ground *S. molle* samples was homogenised with 10 mL of methanol (80:20 (*v*/*v*)) containing 1% trifluoroacetic acid (TFA) using an E-UC13-HD-D Eins Sci (South Africa) ultrasonic bath. This was accomplished at 300 W and 35 °C for 30 min. Constant agitation on a shaker for 24 h then followed. To remove debris, the sample was vortexed and centrifuged (Hermle Z160M, Hermle LaborTechnik, Wehinger, Germany) at 5000× *g*/10 min. The supernatant was then filtered through a 0.22 μm Millipore filter and stored at −20 °C for UPLC-PDA-MS analysis and for the quantification of phytochemicals.

### 2.4. UPLC-QTOF-MS Profiling and Quantification 

The experiment was conducted on a Waters Synapt G2 Quadrupole Time-of-Flight (QTOF) coupled to a Waters Acquity Ultra-Performance Liquid Chromatograph (UPLC) using the methods of [33,34]. Briefly, the column eluate was first detected by a Photodiode Array (PDA) prior to going through the MS, thus allowing for the simultaneous collection of MS and UV spectra. In negative mode, electrospray ionisation was used with a desolvation gas at 650 L/h and 275 °C respectively and a cone voltage of 15 V. The rest of the mass spectrometry settings were optimised for the best sensitivity and best resolution. In MSE or resolution mode, data were obtained by scanning from 150–1500 *m*/*z*. Two channels of MS data were obtained in MSE mode, i.e., at 4 V (low collision energy) and 40–100 V (collision energy ramp) to acquire fragmentation data as well. For accurate mass determination, Leucine enkaphalin was used as a reference mass/ lock mass while sodium formate was used to calibrate the instrument. To achieve separation, the Waters HSS T3 instrument was set at 2.1 × 100 mm, 1.7 μm column. The mobile phase comprised of 0.1% formic acid as solvent (B) as well as 0.1% formic acid containing acetonitrile as solvent A in a combined volume of 2 μL. In a linear way, the gradient commenced at 100% for a minute with solvent A and shifted to 28% for solvent B over 22 min. Consequently, the gradient progressed in solvent B to 40% over 50 s then a wash step of 1.5 min at 100% B, then re-equilibration to the original conditions was instituted for 4 min. Column temperature was kept constant at 55 °C while the flow rate was maintained at 0.3 mL/min. By injecting a range of catechin standards between 0.5 and 100 mg/L catechin, the metabolites were quantified in a relative manner against a calibration curve. The data were processed using MSDIAL and MSFINDER (RIKEN Center for Sustainable Resource Science: Metabolome Informatics Research Team, Kanagawa, Japan).

### 2.5. Chemometric Data Analysis 

The UPLC-QTOF-MS data of *S. molle* SH, DH, and H berries were analysed by Principal Component Analysis (PCA) to identify potential discriminate variables. Using the Markerlynx v4.1., alignment and peak detection and raw data filtering were conducted. A mass range of 100–1000 Da, 5–21 min retention time as well as 50 mDa tolerance time were used as parameters. In addition, 0.4 min retention time tolerance, a 500-intensity threshold/ counts of collection parameters, and a noise elimination level of 1.00 were all set. SIMCA P+ (13.0) software (Umetrics, Umeå, Sweden) was used to determine *m*/*z* data pair and retention time for each peak. The data were then used to determine dataset relationships and various chemometric model constructions.

## 3. Results

### 3.1. Phenolic Acids and/or Derivatives 

As shown in Table 1, MS analysis allowed for the detection and identification of seven phenolic acids and/ or their derivatives in the methanolic extracts of DH, SH, and H *S. molle* berries. A gallic acid and two gallic acid derivatives are phenolic acids that were detected and quantified in *S. molle* berries, while phenolic acid esters including 3-caffeoylquinic acid and feruloyltartaric A and B were also detected. In addition, gentisic acid 5-O-glucoside, a phenolic acid glucoside was also detected although this was undetectable in the dehulled seeds. The dominant phenolic acid was the ester feruloyltartaric A (Figure 2) which recorded 295.7 and 1613.1 mg/kg in SH and H samples respectively. These results show that the husks contained significantly higher phenolic acids in comparison with hull containing seeds and the dehulled seeds in the order H > SH > DH. 

### 3.2. Flavonoids 

MS analysis allowed for the detection and identification of seven flavonols from the flavonoid class of secondary metabolites, in methanolic extracts of S. molle dehulled seeds, hulls and hull containing seeds (Table 1). Similar to phenolic acids, the results also indicate that more compounds were extracted from the hulls while the hull containing seeds and dehulled seeds possessed less compounds respectively. From the nine flavonols that were quantified in the husks, the compounds quantitatively decreased in the order quercetin 3-O-glucuronide > catechin > isorhamnetin galactoside > isorhamnetin glucoside > catechin 3-O-gallate > quercetin 3-(2-galloylglucoside) > pentahydroxyflavanone > quercitrin > quercetin 3-lathyroside. In the dehulled seeds, catechin and quercetin 3-O-glucuronide are the only flavonols that could be quantified. The compounds respectively ranged between 1.8 and 267.5/kg in quercitrin and quercetin 3-O-glucuronide in the seeds + husks. Quercetin 3-O-glucuronide (Figure 3) and catechin were conclusively the most dominant flavonoids in the present study.

### 3.3. Tannins 

As shown in Table 1, fourteen gallotannins, a subclass of hydrolisable tannins were detected and quantified in the husks of S. molle and decreased in the order: trigalloyl glucose > digalloyl quinic acid > galloylsalidroside > pentagalloyl-b-D-glucose > β-glucogallin > pistafolin > eucaglobulin > galloylquinic acid B > galloylquinic acid A > gallic acid glycoside > digalloylshikimic acid A > digalloyl quinic acid > digalloylshikimic acid B. In the seed + husks, the gallotannins ranged between 1.1 and 146.6 mg/kg in pentagalloyl-b-D-glucose and digalloyl quinic acid respectively. In the dehulled seeds, digalloyl quinic acid (21.8 mg/kg), β-glucogallin A (14.8 mg/kg), and digalloyl quinic acid (2 mg/kg) are the only gallotannins that were detected and quantified. Digalloylshikimic acid B (Figure 4) as well as digalloyl quinic acid were therefore the dominant tannins in the present study.

Two compounds possessing the molecular formula C_17_H_30_O_9_ were also revealed although they could not be identified. Like phenolic acids, flavonols and gallotannins, the two compounds decreased in the order H > SH > DH. Also, an organic acid (quinic acid derivative) was detected and was found to be 9, 20 and 31 mg/kg in SH, DH, and H samples respectively. The dehulled samples of *S. molle* berries were surprisingly higher in quinic acid derivative than the seed + husk samples. 

## 4. Discussion

### 4.1. Phenolic Acids and/or Derivatives 

Acid and/or alkaline hydrolysis of tannins gives rise to gallic acids (GA), a trihydroxybenzoic compound of low molecular weight/ phenolic acid [38]. GA is naturally occurring and widely distributed in plants and can exist either as part of tannins or freely [38]. The diversity and antioxidant potential of GA and associated derivates are well known in the plant kingdom. Besides these antioxidant activities, GA derivatives have been reported to induce apoptosis in cancer cells, scavenge free radicals, signal pathway interference involving oxygen free radicals and Ca^2+^, as well as squalene epoxidase inhibition [39]. Also, some antitumoral, antiviral, antibacterial, antifungal, and anticancer activities have been linked with GA derivatives and esters, including activities in aging, neurodegenerative disorders, and cardiovascular diseases [38]. In addition, according to the previous authors, these activities are thought to be linked to the hydrophobic moiety of GA. Of particular interest is the earlier cytotoxicity study of [40], which showed that GA induced cell death in human promyelocytic leukaemia, mouse lymphoid neoplasm, human epithelial carcinoma, rat hepatoma, human hepatoma, and human epidermoid carcinoma cells, but with relatively high selectivity. Therefore, the presence of GA, its esters, and derivatives in *S. molle* berries offer insights into the potential of these berries in curing cancer and related diseases. However, besides the biological and pharmacological activities, esters of gallic acid prevent oxidation in foods while some cosmetic uses such as eye makeup, skin care, and hair shampoo among others have also been reported [41]. 

Some of the QA derivatives that were found in the present study are galloylquinic acid A and B (Gallotannins) as well as 3-caffeoylquinic acid (phenolic acid). Feruloyltartaric, a cinnamate ester obtained by formal condensation of the carboxy group of ferulic acid with one of the hydroxy groups of tartaric acid [35] is the most abundant compound in this study. The antidiabetic, antihypertensive, antioxidant, and anticancer qualities of ferulic acid are well known [42]. However, according to the previous authors, the bioavailability of this promising compound is low. But ferulic acid bioavailability is known to be improved by feruloyl esterases [42]. The presence of feruloyltartaric could therefore also conceivably help improve the bioavailability of ferulic acid and help improve the health promoting qualities of *S. molle*.

In comparison, the work of [43] was able to profile and quantify two phenolic acids including gallic acid and vanillic acid in two separate regions of Tunisia. Our work was able to profile and quantify three phenolic acids, 13 gallotannins, a phenolic acid glucoside and, three phenolic acid esters. Quantitatively, the Tunisian berries show more elevated concentrations. The previous authors were also able to show that *S. molle* berries quantitatively contained more phytochemical constituents in comparison with *S. terebinthifolius*, a close member of the species.

### 4.2. Flavonoids 

The antioxidant, anti-inflammatory, and anti-carcinogenic role of flavonoids are well reported in literature. The results of the present study showed that flavonoid-3-O-glycosides dominated the flavonoid class and included isorhamnetin galactoside, isorhamnetin glucoside, quercetin 3-lathyroside, quercetin 3-(2-galloylglucoside), and quercitrin. The antioxidant, anti-inflammatory, antidiabetic, anticancer, anti-HIV activity, anti-obesity, anti-rotavirus activity, anti-influenza viral activity, anti-mayaro viral activity, anti-tumour, anti-coagulant, and anti-platelet activity of flavonoids have been widely reported [44,45,46,47,48]. The husks of *S. molle* berries were abundant in catechin, indicating the potential of the berries in human health.

In another study, the work of [43] showed only five flavonoids from *S. molle* and *S. terebinthifolius* berries. These authors reported catechin, epicatechin, rutin, luteolin, and kaempherol in their samples. Although our work reports nine flavonoids, only catechin was found in common between our works.

Other studies conducted so far show that *S. molle* essential oil extracts of the berries are different from the current methanolic extracts, for example the results of the studies of [16,19,27,29,30]. These authors reported different polyphenols from our findings. However, the alcoholic extracts of [31,32] compare well with the present results. These results further indicate that different extraction methods and instrumentation potentially yield different compounds. The essential oils in these previous studies were characterised by a high composition of α-pinene, the major monoterpene hydrocarbon class compound. The previous authors found α-pinene to dominate the essential oil compounds while phenolic acids dominated the methanolic extracts. Our findings showed that phenolic acids and their derivatives dominated the secondary metabolites while flavonoids were minimal. 

Literature is awash with evidence that secondary metabolites are more concentrated in the peels/ or husks of fruits or berries [49,50]. It is therefore not surprising that the current results profiled and quantified more phenolic acids and/or their derivatives, tannins, and flavonoids in the husks in comparison to the dehulled seeds. As far as we know, the present study reports for the first time the secondary metabolites profile of *S. molle* berries grown in South Africa and the differences among the husks, dehulled seeds and seed containing husks in *S. molle* berries. A Korean study earlier quantified piperine, gallic acid, protocatechuic acid, epicatechin, and p-coumaric acid in the berries [32]. The Tunisian study of [31] also quantified these metabolites and many others in the berries of *S. molle* although these authors’ values are lower than our findings. We therefore conclude based on our results that the husks are very rich in both phenolic acids and/ or their derivatives, tannins, and flavonoids.

### 4.3. Tannins 

The antioxidant, antitumor, and anticancer activities are among activities too many to mention that have been associated with catechin and its derivatives [51]. Tannins are particularly abundant in pulses, mostly red coloured beans, and are well known for their astringency [52,53]. Their molecular weight ranges from 500 to over 3000 and their ability to precipitate proteins, amino acids, and alkaloids have also been reported [52,53]. Anti-inflammatory studies indicate that tannins can control enteritis, bowel disorders, esophagitis, diarrhea, all forms of gastritis, kidneys, dysentery, fatigue, haemorrhaging, skin ulcers, and sore throats among others [54,55]. Furthermore, tannins have been shown to heal wounds and/or burns and stop bleeding simultaneously [55]. They heal wounds by forming a protective layer over the wound, and this consequently stops infections [55]. The anti-tumour, antibacterial, antiviral, and antiparasitic properties of tannins have also been reported [56]. *S. molle* berries are rich in gallotannins, a tannin subclass of hydrolysable tannins and as such potentially possess some of these health benefits. Heating tannins with hydrochloric or sulfuric acid may result in gallic or ellagic acids [57]. These changes from hydrolysable tannins to gallic or ellagic acid conceivably take place in the gut during digestion and thus some health benefits that are linked to these products of digestion may be realised. Gallic acid, a possible derivative of hydrolysable tannins is well known for inhibiting HIV replication [58]. *S. molle* is also rich in gallic or ellagic acids and derivatives as shown in Table 1 and as such could provide some health-related benefits.

Some of the tannins have not been widely studied and very little information exists on their role in pharmacology. For example, the study of [58] is the only study we found that reports on digalloylshikimic acid, while the studies of [59,60,61] report on pistafolin and showed some activity against ROS. The biological activities, mechanism of action, bioavailability among other pharmacological activities of these and other compounds need further investigation to reveal their full potential. These compounds are also abundant in *S. molle* berries as indicated in our results. However, phenolic acids in *S. molle* berries are promising especially in the husks. Table 1 further shows that most of the compounds that were detected are somewhat related, one being a product of hydrolysis of the other. Further studies are needed to further elucidate these relationships and possibly show how these interact in the human body as well as their bioavailability for nutrition and health. 

Eucaglobulin, a gallotannin and one of the compounds that to the best of our knowledge is being reported for the first time in *S. molle* and possesses some anti-melanogenic and anti-inflammatory activity [62]. This compound has previously been reported in *Eucalyptus globulus* leaves and other Eucalyptus species [63].

Quinic acid (QA), a cyclohexanecarboxylic acid is an organic acid that is diverse in the plant kingdom [64]. According to the previous authors, although QA is naturally formed from dehydroquinic acid, which is an intermediate in the shikimic pathway, it has been synthetically hydrolysed from chlorogenic acid. This compound is dominant in the hulls of *S. molle* berries of the current study. In pharmaceuticals, QA is used as an astringent or a chiral substance and is an oseltamivir building block, an antiviral medicine that is used to treat infleunza A and B [65]. But biological activity studies have revealed the antioxidant, antidiabetic, radioprotective, and anti-neuroinflammatory activities of QA and its derivatives [66]. 

### 4.4. Multivariate Analysis

Using an untargeted metabolomics approach, phytochemical differences between SH, DH, and H of *S. molle* berries were broadly evaluated and characterised. The use of metabolomics as a tool for complex plant sample, feed product, product adulteration, cultivar variations and food quality elucidation studies are well documented [67]. In the present study, visualizing UPLC profiles of extracts from *S. molle* DH, SH, and H berries with varying phytochemical profiles and contents of individual metabolites with varying susceptibility led to the observation of variable but non complicated patterns. The metabolites shown in Table 1 were subjected to PCA using MetaboAnalyst 5.0 software for computing metabolomics data. Using this software, differences among DH, SH, and H of the berries were computed and profiled following the methods of [68] and following the workflow for untargeted metabolomics. The main principal component (PC); PC1 and PC2 accountable for 91 and 7.1% of the variance respectively as shown in Figure 5. The effectiveness of PCA in studies involving classification of species and when plotting exploratory analysis has been demonstrated [69]. PCA scores of the present study have unfolded a lot of data on polyphenolic compounds in *S. molle* SH, DH, and H berries. Three groups of populations have been separated and categorised by the two score plots shown in Figure 1. All three groups were placed separately with SH and H components towards the right and somewhat close to each other. DH seeds were placed far away from the H, showing the strong influence of the husks in determining the presence of secondary metabolites. The SH samples are close to either DH or H samples but not very close to either, showing that husks have an influence on the secondary metabolite profiles and quantities. Pigmented colours in fruits and vegetables are closely related to secondary metabolite profiles and quantities. The dehulled seeds of *S. molle* appear brownish in colour, the pink to reddish pigment is confined to the hulls. Therefore, removing the hulls conceivably reduces the secondary metabolites. It is also conceivable that the high oil contents of the seeds somewhat interfere with the presence of secondary metabolites. This is because the metabolites are drastically reduced when the seeds and husks are combined. 

The intensity of the various untargeted polyphenolic compounds in *S. molle* SH, DH and H berries are shown in the heat map on Figure 6a. Included in this heat map are the clustergrams where the raw data represented the secondary metabolite compositions relating to whether the *S. molle* berries were dehulled, hulled or both. The colour block represented the intensity of the treatments of secondary metabolite concentrations. The lower concentrations were represented by the colour blue while the higher concentrations were represented by the colour red as shown in Figure 6b. Therefore, based on the heat map, husks had higher polyphenolic concentrations in comparison to lower concentrations in SH and DH samples respectively.

## 5. Conclusions

The results of the current study indicate that *S. molle* berries are a rich source of secondary metabolites. Thirteen gallotannins, three phenolic acids, a phenolic acid glucoside, three phenolic acid esters, an organic acid, a gallotannin derivative, and nine flavonoids were profiled and further quantified. The present study revealed that feruloyl tartaric A, a phenolic acid ester was the dominant phenolic acid while quercetin 3-O-glucuronide and catechin were the major flavonoids, and digalloylshikimic acid B as well as digalloyl quinic acid were the dominant tannins. The husks possessed more metabolites than the seeds + husks and dehulled seeds respectively. Although the use of the husks alone for culinary purposes may prove difficult, perhaps, the use of the whole berry would be ideal. However, to extract more secondary metabolites for pharmaceutical or food industry purposes, it would be ideal to extract from the hulls. Although *S. molle* is considered invasive in South Africa, both culinary and pharmaceutical applications can be derived from the ripe and mature berries, making this plant potentially an important nutraceutical plant. However, further studies need to be conducted to further elucidate the pharmacological properties and biological activities that can be linked to the reported polyphenolic compounds. The berries of these trees could also potentially provide sources of income in poor and marginalised communities if commercial linked applications can be harnessed. South African communities also need to learn about the existing culinary uses of these berries. 

## Figures and Tables

**Figure 1 foods-11-01376-f001:**
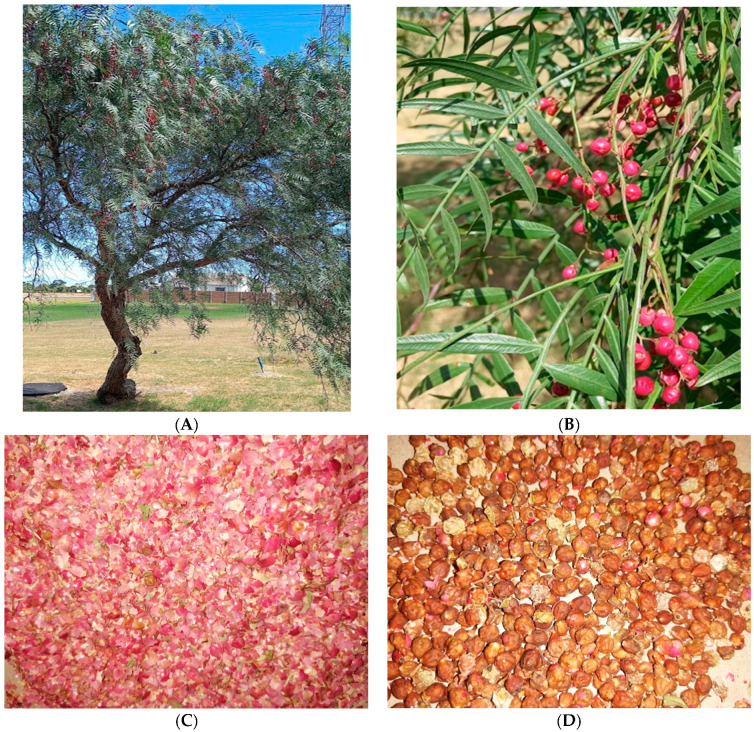
(**A**)—*S. molle* tree; (**B**)—Ripe *S. molle* berries on the tree branches; (**C**)—Husks of *S. molle* berries; (**D**)—Dehulled *S. molle* berries.

**Figure 2 foods-11-01376-f002:**
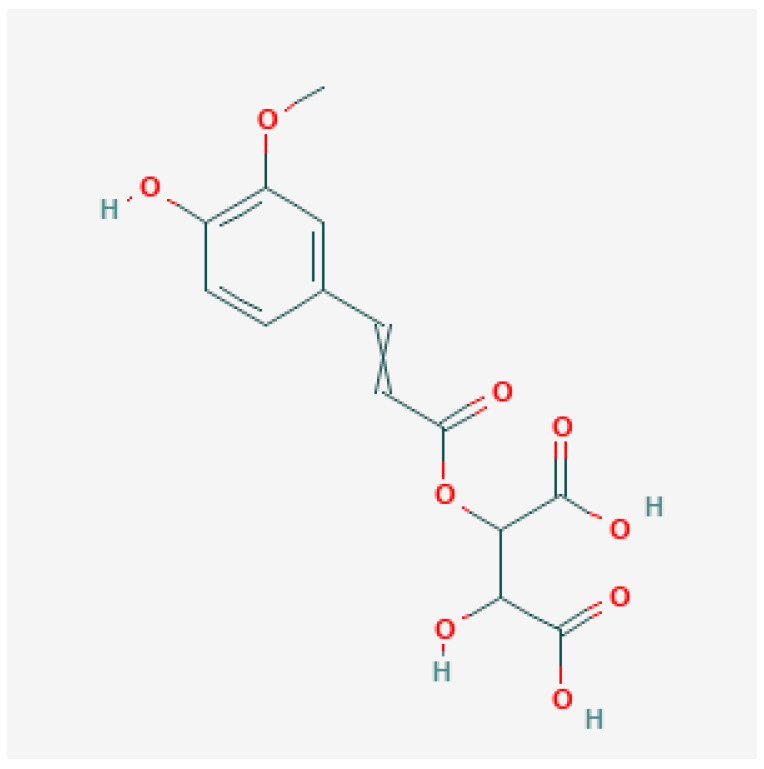
Feruloyltartaric acid chemical structure [35].

**Figure 3 foods-11-01376-f003:**
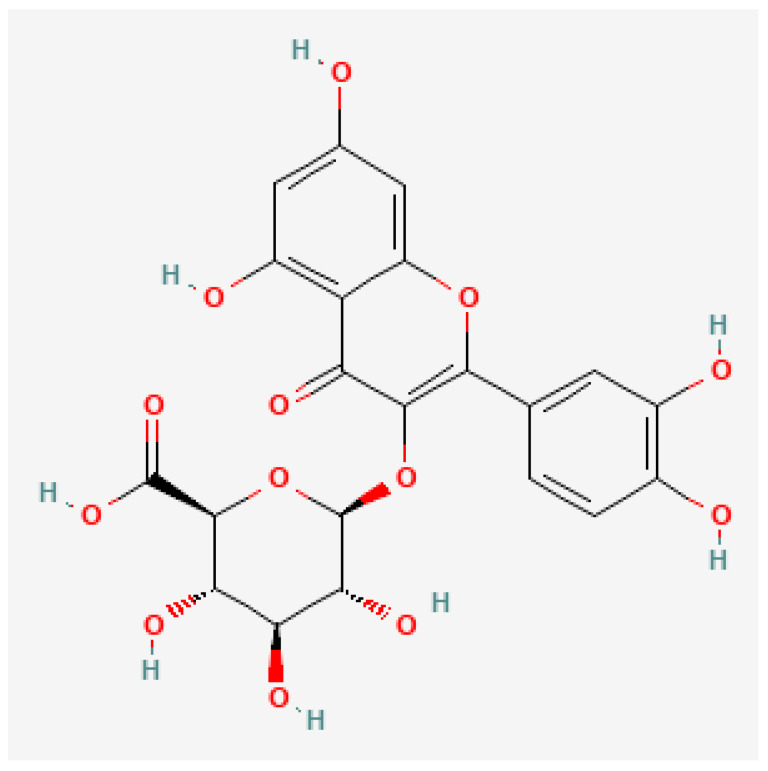
Quercetin 3-O-glucuronide chemical structure [36].

**Figure 4 foods-11-01376-f004:**
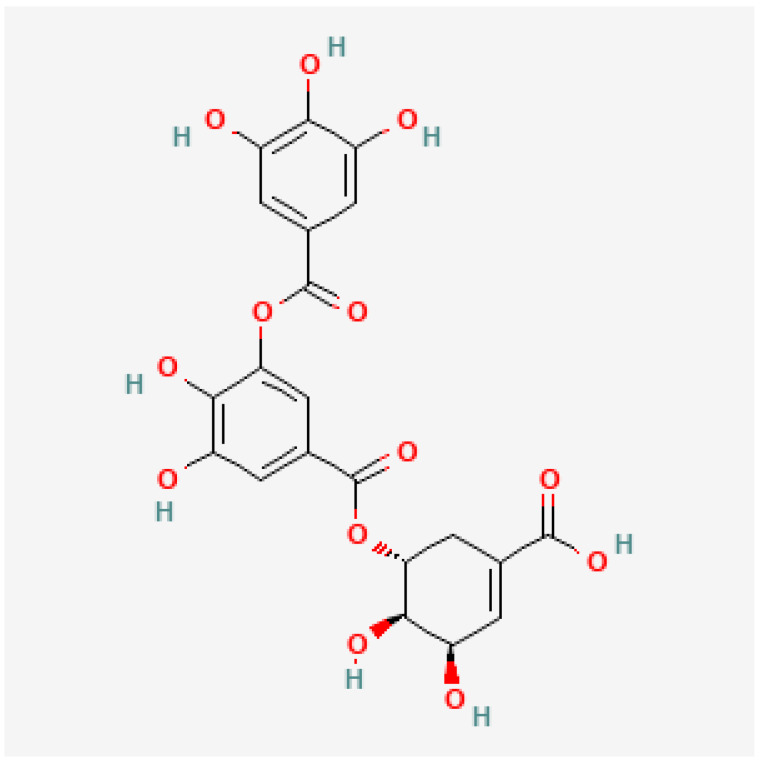
Digalloylshikimic acid chemical structure [37].

**Figure 5 foods-11-01376-f005:**
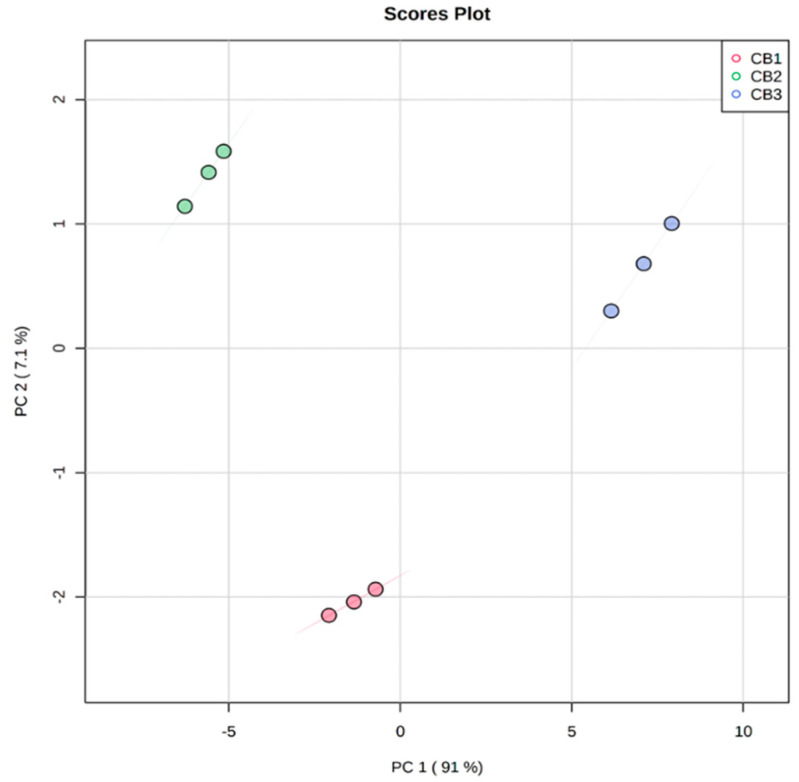
Principal component analysis Score plot of PC1 Vs PC2 scores of *S. molle* Seeds + Husks (CB1), Dehulled Seeds (CB2) and Husks (CB3) analysed via UPLC-QTOF-MS.

**Figure 6 foods-11-01376-f006:**
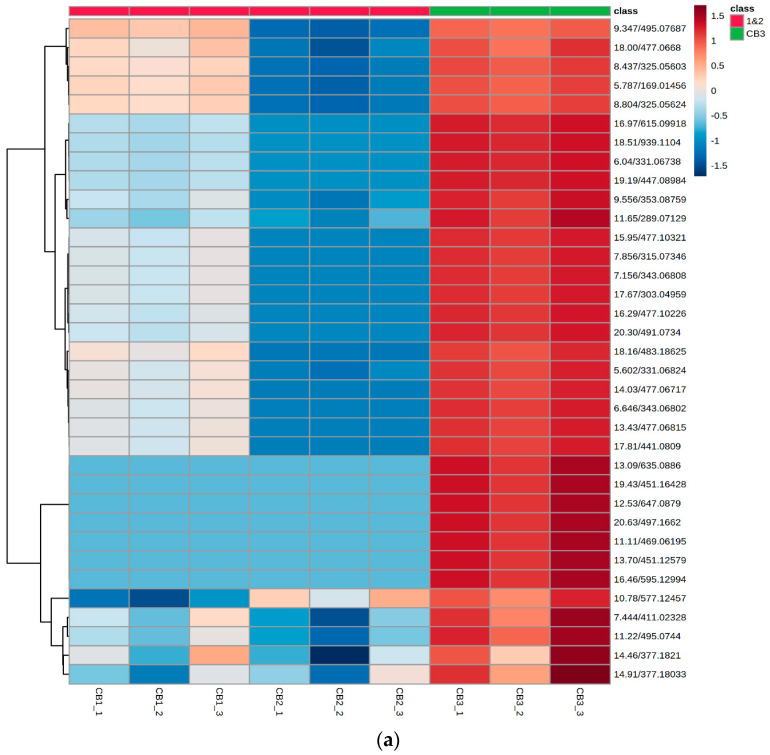
(**a**): Heat map of *S. molle* Seeds + Husks (CB1), Dehulled Seeds (CB2), and Husks (CB3) secondary metabolites analysed via UPLC-QTOF-MS. The numbers on the right represent the RT and *m*/*z* [M-H]^−^ shown on Table 1. (**b**): Heat map of *S. molle* Seeds + Husks (CB1), Dehulled Seeds (CB2), and Husks (CB3) secondary metabolites analysed via UPLC-QTOF-MS.

**Table 1 foods-11-01376-t001:** Secondary metabolites in *Schinus molle* berries (acidified methanolic extract) by LC-MS in negative ionization mode.

RT	UV Max	*m*/*z* [M-H]^−^	Tentative Identification	Formula	Class	SH (mg/kg *)	DH (mg/kg *)	H (mg/kg *)
5.602	229.6	331.06824	β-Glucogallin A	C_13_H_16_O_10_	Gallotannin	76.0	14.8	443.6
5.787	271.6	169.01456	Gallic acid	C_7_H_6_O_5_	Phenolic acid	62.7	4.9	252.3
6.04	273.6	331.06738	β-Glucogallin	C_13_H_16_O_10_	Gallotannin	2.1	nd	147.8
6.646	268.6	343.06802	Galloyl quinic acid A	C_14_H_16_O_10_	Gallotannin	30.8	nd	265.8
7.156	280.6	343.06808	Galloyl quinic acid B	C_14_H_16_O_10_	Gallotannin	23.9	nd	251.9
7.444	255.6	411.02328	Gallic acid derivative	C_9_H_16_O_18_	Phenolic acid	99.0	77.0	166.4
7.856	231.6	315.07346	Gentisic acid 5-O-glucoside	C_13_H_16_O_9_	Phenolic acid glucoside	12.2	nd	130.3
8.437	261.6	325.05603	Feruloyl tartaric A	C_14_H_14_O_9_	Phenolic acid ester	295.7	20.2	1613.1
8.804	267.6	325.05624	Feruloyl tartaric B	C_14_H_14_O_9_	Phenolic acid ester	157.0	7.2	908.8
9.347	272.6	495.07687	Digalloyl quinic acid	C_35_H_28_O_22_	Gallotannin	146.6	2.0	561.6
9.556	229.6	353.08759	3-caffeoylquinic acid	C_16_H_18_O_9_	Phenolic acid ester	95.1	36.4	543.9
10.781	274.6	577.12457	Quinic acid derivative	C_26_H_26_O_15_	organic acid	9.0	20.5	31.0
11.11	275.6	469.06195	Gallic acid derivative	C_19_H_18_O_14_	Phenolic acid	nd	nd	89.7
11.227	272.6	495.0744	Digalloyl quinic acid	C_35_H_28_O_22_	Gallotannin	28.5	21.8	60.8
12.537	237.6	647.0879	Pistafolin	C_28_H_24_O_18_	Gallotannin	nd	nd	147.9
13.094	237.6	635.0886	Trigalloyl glucose	C_27_H_24_O_18_	Gallotannin	nd	nd	48.1
13.432	235.6	477.06815	Digalloylshikimic acid A	C_21_H_18_O_13_	Gallotannin	65.8	nd	524.6
13.705	235.6	451.12579	Galloylsalidroside	C_21_H_24_O_11_	Gallotannin	nd	nd	88.1
14.034	274.6	477.06717	Digalloylshikimic acid B	C_21_H_18_O_13_	Gallotannin	102.4	nd	664.8
14.464	235.6	377.1821	Unidentified	C_17_H_30_O_9_		121.4	103.0	155.4
14.919	235.6	377.18033	Unidentified	C_17_H_30_O_9_		233.3	240.7	378.0
18.168	237.6	483.18625	Gallic acid glycoside	C_23_H_32_O_11_	Gallotannin	108.1	nd	469.1
18.513	237.6	939.1104	Pentagalloyl-β-D-glucose	C_41_H_32_O_26_	Gallotannin	1.1	nd	96.3
20.639	276.6	497.1662	Eucaglobulin	C_23_H_30_O_12_	Gallotannins derivative	nd	nd	174.5
15.952	235.6	477.10321	Isorhamnetin galactoside	C_22_H_22_O_12_	Flavonol	38.4	nd	430.1
16.299	235.6	477.10226	Isorhamnetin glucoside	C_22_H_22_O_12_	Flavonol	27.4	nd	377.3
16.461	235.6	595.12994	Quercetin 3-lathyroside	C_26_H_28_O_16_	Flavonol	nd	nd	124.0
16.978	237.6	615.09918	Quercetin 3-(2-galloylglucoside)	C_28_H_24_O_16_	Flavonol	4.2	nd	224.1
17.67	235.6	303.04959	Pentahydroxyflavanone	C_15_H_12_O_7_	Flavonol	16.5	nd	172.6
18.004	254.6	477.0668	Quercetin 3-O-glucuronide	C_21_H_18_O_13_	Flavonol	267.5	73.4	564.3
19.198	237.6	447.08984	Quercitrin	C_21_H_20_O_11_	Flavonol	1.8	nd	126.3
17.812	234.6	441.0809	Catechin 3-O-gallate	C_22_H_18_O_10_	Flavanol	41.7	nd	326.8
11.651	229.6	289.07129	Catechin	C_15_H_14_O_6_	Flavanol	122.0	80.4	551.9

**Where:** SH—Seed + Hulls; DH—Dehulled Seeds; H—Husks; nd—not detectable; * relative to catechin calibration curve; RT–retention Time; *m*/*z*—Mass-to-Charge Ratio; UV max—Maximum UV.

## Data Availability

Data is contained within the article.

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
