# Peer review of "Composition of Phenolic Compounds in South African *Schinus molle* L. Berries"

_foods, 2022, doi:10.3390/foods11101376_

Round 1
Reviewer 1 Report
The purpose of the study by Callistus Bvenura and Learnmore Kambizi was to investigate the polyphenolic compound variations in South African Schinus mole L. berries. Various studies have indicated that phenolic compounds have drawn the attention of many researchers due to their various biological effects.
In the Introduction, the authors also inform about the multidirectional pro-health effects of Schinus molle, mainly the activity of essential oils. Less information relates to the activity of phenolic compounds present in the fruit of this plant. Therefore, taking up the proposed topic is justified. Nevertheless, the presented manuscript requires an in-depth correction. There are many issues that need clarification and improvement.
- The title of the manuscript should be corrected as the dominant quantitative hydroxybenzoic acids are not polyphenols but phenolic acids. Polyphenolsare an organic compounds characterized by multiples of phenol units ( at least two). On the other hand, catechin and catechin-3-O-gallate were classified by the authors as tannins. They are flavan-3-ol monomers, which are the building blocks of condensed tannins which are oligomers and polymers of catechin units joined by carbon-carbon bonds.
- Abstract - if the goal was to determine the non-polar metabolome of Schinus molle why was methanol used for their extraction, which is a polar protic solvent.
- Section 2.1. - the authors mention "tetrafluoro acetic acid", and in section 2.3. they use trifluoroacetic acid for extraction. Please indicate at what stage of research tetrafluoroacetic acid was used.
- Section 2.2. - please provide a pictorial drawing of Schinus molle berry and the separated fractions. The description included in the work is ambiguous.
- Section 2.2. and 2.3. incorrect temperature unit notation.
- Section 2.2. - please explain the wording “Where necessary, the samples were dried ….”. What criterion was the indication for the drying to be carried out? Was the dry weight of the resulting droughts determined?
- Section 3 - requires supplementing with the obtained chromatograms, providing in Table 1 the value of the maximum wavelength recorded in the absorption spectrum and the data (m/z) obtained after fragmentation. The identification presented in Table 1 should be better described and documented in the results.
- Table 2 – gentesic acid should be gentisic acid; Pentagalloyl-beta-D-glucose is a gallotannin; catechin and catechin-3-O-gallate are flavanols. Whereas, quercetin, isorhamnetin and their derivatives are among the flavonols (which of course are a subgroup of flavonoids). Please replace "deriv" with the full name. No description after table 1 of the abbreviation “nd”.
- Line 178 - what was the detection limit for the compounds listed, please specify.
- Line 196 - which means the statement of “triphenolic compound” in relation to gallic acid.
- Discussion - the authors focused on listing the biological activity of the identified compounds. There is no discussion regarding, for example, the influence of the food matrix on their activity, their actual bioavailability. The presence of other ingredients of Schinus molle berry also influences the activity of phenolic compounds, which the authors do not take into account in the discussion. There is also no reference to other works describing the identification of phenolic compounds, if not in Schinus molle then in plants of the same family.
- Conclusions are very general and does not refer to specific own data presented in the paper.
Reviewer 2 Report
The manuscript titled “Polyphenolic compound variations in South African Schinus molle L. berries” is related to the determination of phenolic profile for hulls, seed with hulls, and seeds of Schinus molle L. berries. There are several major issues:
1. Discussion part is generally oriented towards explanation of bioactivities for individual phenolics which were reported by other studies. Yet, the authors did not perform any test for biological activity (antioxidant, antimicrobial, antiproliferative etc) for Schinus molle L. phenolics extracts.
2. Extractions without hydrolysis are not performed which is particularly important since studied plant material is not widely explored, thus this information could be relevant for the scientific community.
Since samples were collected in May 2021, bioactivity of obtained samples probably deteriorated and it is not possible to perform any additional tests . But I would like to stress that the phenolic profile of Schinus molle L. was not widely studied thus manuscript has a certain value.
Below are listed other comments which could improve the quality of the manuscript:
Line 28 – Please insert first the full term of WHO
Lines 35-37 – Please rephrase the sentence.
Line 39-41 – polyphenolic compounds; increased physical activity
Line 55-57 – Please add the references where it was reported.
Lines 60-61 – Is it California?
2.3 – Please add information about ultrasound and centrifuge (model, producers, and country of production). Also, add temperature of ultrasound extraction and ultrasound power.
Please explain why also extraction procedure without TFA was not applied? It is hard to conclude which phenolics are naturally present in plant material and which phenolics are products of hydrolysis. Extraction without hydrolysis is particularly important since studied plant material is not widely explored, thus this information could be relevant for the scientific community.
Lines 212-213 - Please add these cosmetic uses.
Lines 270-274 - The straightforward comparison with reference 30 in terms of concentration of individual phenolics is not possible since authors in ref 30 used different phenolics to calibrate the equipment. The discussion should be oriented towards qualitative profile and reporting the most significant compounds, while their exact concentrations are only approximate.
Also, ref 31 and all studies where phenolic acids were detected for Schinus molle berries should be mentioned in section 4.1. The same goes for sections 4.2 and 4.3.
All works related to the phenolic profile of Schinus molle should be included, one of the missing works is:
Tlili, N., Yahia, Y., Feriani, A., Labidi, A., Ghazouani, L., Nasri, N., ... & Khaldi, A. (2018). Schinus terebinthifolius vs Schinus molle: a comparative study of the effect of species and location on the phytochemical content of fruits. Industrial Crops and Products, 122, 559-565.
Lines 280-290 – Which tannins are involved in discussion. This part is to general, bioactive compounds should be specified
Figure 1. Please explain what does 3 dots for each fraction represents. Three repetitions of extraction? Three groups of phenolics? Furthermore, is figure 1 really necessary? It is obvious from Table 1 that husk has the most of the phenolics.
Figure 2a Similar question, please explain what does CB1_1, CB1_2 and CB1_3 stands for? Also, legend for classes seems to be incorrect.
Reviewer 3 Report
The presented manuscript foods-1605865 entitled "Polyphenolic compound variations in South African Schinus molle L. berries" is dedicated to chromatographic quantification of phenolic compounds in Schinus molle tree.
Authors have shown that the contents of phenolic acids, flavonoids and tannins change in wide ranges for different samples. It is not clear, how many samples were analysed? What the difference between them?
I am afraid, that I miss the aim of the study. The presented results of analysis can be as significant part of the paper, but additional experiments (e.g., antioxidant properties evaluation) are needed.
Text corrections are needed:
- Italic font "Schinus molle" in line 49, 75 ?
- Please, format the References according to journal template:
line 72: "[12, 13, 14]. The studies of [15], [16], [17] and [18]",
also lines 74, 75, 228, 249, 252 etc.
- Probably, aqueous methanol? - line 106
- "centrifuged at 5000g/ 10 min" line 109 - need to be corrected
- "molleberries" line 154 - missed space
- Format of Tables different as in journal template.
- upper script "m−1" line 91, "Ca2+" line 202
- Missing Table 2? - line 308
Reviewer 4 Report
Bvenura and Kambizi have investigated the variations of polyphenolic compounds in South African Schinus molle L. berries. This article is well-written and contains some methodically obtained quantitative data. However, there are several key information missing which should be provided along with some added discussion, as pointed out below:
- Section 2.1 – the details of state, city and country should be added to Sigma Chemicals if it is from USA. If not, both city and country should be provided.
- Section 2.3 – Why TFA was added to methanol for sample extraction? Please brief the role of TFA.
- Section 2.4 – How the quantitation of phenolic acids, flavonoids and tannins are made by only using catechin standards? The authors should address this issue.
- L127-128 – “0.1% formic acid as solvent (A) as well as 0.1% formic acid containing acetonitrile as solvent B” should be corrected as “0.1% formic acid as solvent (A) and 0.1% formic acid containing acetonitrile as solvent (B)”.
- Section 2 – As the authors have adopted a reported analytical method from [32] and [33], the details of the validation parameters such as LOD, LOQ, recovery and precision data should be included to show the validity of the employed analytical method.
- It is important to include the chemical structures of dominant major phenolic acids, flavonoids and tannins quantified to be present in molle.
- Section 2 – a new sub-section 2.6 “Statistical analysis” should be included detailing the statistical method and software used in this study.
- Figures 1 and 2 – there is no discussion on the findings from the PCA and heat map analysis. The authors should include two sizable paragraphs for one each for describing the observations/outcomes from these analyses.
- Figure 2a and 2b – these two figures should be provided with a common caption as they are components of a single figure.
- Table 1 – the abbreviations such as RT and m/z should be abbreviated as retention time and mass-to-charge ratio, respectively. Also, “Tentative ID” should be modified to “Tentative identification”.
- All the abbreviations should be checked for providing the full form in the first instance and abbreviating thereafter.
- All the values and units should be separated by a space.
- The hour, minutes and seconds should be abbreviated as h, min and s.
- In all the places degree symbol should be superscripted in degree centigrade as °.
Round 2
Reviewer 1 Report
The reviewers' suggestions were taken into account in the revised manuscript. Unfortunately, the content of the manuscript still requires minor corrections.
1. The title is still hard to read. Please consider, for example, Composition of phenolic compounds in South African Schinus molle L. berries.
2. Please change the spelling of ferruoyl tartaric to ferruoyltartaric, galloyl quinic to galloylquinic in the manuscript and in Table 1
3. Unify UPLC-QTOF-MS (line 213) or UPLC-qTOF-MS (line 10, 652, 655)
4. Line 253 and 666 - should be nine flavonoids (or seven flavonols and two flavanols) instead of nine flavonols
5. Line 254,255 the names 3-o-glucuronide and 3-o-gallate should be written as 3-O-glucuronide and 3-O-gallate
6. Line 327 S.molle should be written in italics
7. Line 351 and 353 - S. molle probably entered 3 times unnecessarily
8. Please write the names of the relationships in the middle of the sentence with lowercase letters instead of capital letters (e.g. line 367-449; 474-475 or 495)
Reviewer 2 Report
After the second evaluation and checking all questions raised by other reviewers I have decided to stay with the previous conclusion. Mainly due to weaknesses in research design e.g. not including bioactivity tests and not considering extraction without TFA. Furthermore, authors referred a lot to technicians overseas when it comes to extraction and UPLC-QTOF-MS analysis (standard issue) which are two main experimental tools of this study. Furthermore, LOQ, LOD, precision, and recovery which other reviewer requested were not provided.
Reviewer 3 Report
The authors have satisfactorily responded to all comments raised by reviewers.
Reviewer 4 Report
The authors have satisfactorily responded to all comments raised by reviewers.
